# Influence of large open-pit mines on the construction and optimization of urban ecological networks: A case study of Fushun City, China

Dongmei Feng, Ge Bai ☉*, Liang Wang

College of Business Administration, Liaoning Technical University, Huludao, Liaoning, China

* 2433293485@qq.com

## Abstract

Under the long-term effect of mineral resource exploitation, especially open-pit mining, ecosystems are severely disturbed. Constructing and optimizing urban ecological networks influenced by open-pit mines based on mine–city coordination helps integrate ecological restoration and the construction of urban ecological environments. We applied an InVEST model to Fushun City to evaluate urban ecosystem services under the influence of large open-pit mines. Twenty-one key patches important for maintaining landscape connectivity were screened as the ecological sources in the network, from which ecological resistance surfaces were constructed by combining the impacts of mines on the environment. Minimum cumulative resistance (MCR) and gravity models were then used to extract and classify ecological corridors favorable to species migration and diffusion. Fushun City had large spatial differences in ecosystem service functions, with high-value areas concentrated in the forest-rich Dongzhou District and the northern Shuncheng District. Under the influence of open-pit mining, the ecosystem service capacity of the region south of the Hunhe River was poor and lacked ecological sources. Urban ecological resistance surfaces reached a maximum in the open-pit mining area, and 210 ecological corridors were estimated using the MCR model, of which 46 were important. Only two corridors crossed the West and East open pit, forming two "ecological fracture surfaces." The Dongzhou and eastern Shuncheng districts had complex network structures and stable ecological environments. In contrast, the central and southern parts of Fushun City lacked ecological corridors owing to the influence of mining pits and gangue mountains, had simple network structures, and low connectivities with other sources. Combined with Fushun City's development plan, we propose that ecological network optimization should add new ecological source sites, reconstruct and repair ecological corridors, and upgrade ecological breakpoints. This study provides reference and basis for ecological network research in mining cities influenced by open-pit mines.

**Data Availability Statement:** The data that support the findings of this study are openly available in public repository "Figshare" at https://doi.org/10.6084/m9.figshare.25390726.v1.

**Funding:** This research was supported by the following grants:(1) Ministry of Science and Technology of the People's Republic of China (No. 2017YFC1503102). (2) The Educational Department of Liaoning Province (No. LJKR0141). (3) Liaoning Technical University (No. 55230010A032). The funders had no role in the study design, data collection and analysis, decision to publish, or manuscript preparation.

**Competing interests:** The authors have declared that no competing interests exist.

## Introduction

Although mineral resources have contributed notable to China's economic and regional development, they have also harmed the ecological security of the region [1,2]. During mining, particularly open-pit mining, vegetation, topsoil, and rock removal occupy and destroy land resources, thereby rapidly aggravating the fragmentation degree of the mining landscape [3–5], reducing regional habitat patch areas [6,7], and destroying the integrity of the mining ecosystem's health and its normal ecological processes [8], as well as negatively affecting the ecological processes of whole cities. Large pits, gangue hills, and geological disasters that occur during long-term open-pit coal mining have hindered or even blocked ecological flow and material circulation in cities [9], which leads to increasingly prominent ecological safety problems in cities [10].

Transforming and developing coal-resource-exhausted cities, as well as constructing ecological civilizations, have higher implications for protecting the ecological environment in mining cities. The ecological restoration of a mine can play a role in governing the mine structure, improving the soil quality, restoring vegetation, and restoring/rebuilding the ecosystem in the mining area [11]. However, after treating a single mining area, the focus should not only be on its internal landscape effect and ecological functions, but also the relationships between restoring the mine ecology and constructing and developing the ecological environment of mining cities to provide ecological security for such cities. The ecological network is a system that organically connects regional ecological resource patches and ecologically weak areas through linear spatial corridors [12] that allow broken ecological patches to connect and form a stable network structure. This structure has positive effects on optimizing landscape patterns and energy flow exchanges [13]. Thus, constructing and optimizing ecological networks are considered to be powerful measures for solving urban ecological environmental problems and ensuring ecological security [14].

Previous studies grounded in landscape ecology theory have investigated the ecological environment at various scales, thereby forming the fundamental research paradigm of ecological networks: ecological source–ecological resistance surfaces–ecological corridors [7,15]. To identify ecological source sites, previous studies have used the attributes of the patches themselves (e.g., nature reserves, large forested areas, and grasslands) [16–19] or have selected ecological indicators, such as ecosystem service capacity [20,21] and ecological sensitivity [22,23] or vulnerability [24]. However, these methods do not adequately consider the connectivity between the patches. Combining ecosystem services with landscape connectivity can improve source site selection [25–27]. To construct resistance surfaces, previous studies have considered natural conditions and the ecological environment (e.g., slope, elevation, vegetation cover, and land use patterns) [18,28]. In addition, some researchers have considered urbanization and human activity, both of which put stress on the ecological environment (e.g., population density[29] and the amount of nighttime light [6,30]), when modifying ecological resistance surfaces to increase the accuracies of the simulated spatial distributions of ecological corridors [31]. However, in the past, the hotspots of ecological network research mainly included urban areas [21,32], arid or semi-arid areas [33,34], agricultural and pastoral intertwined zones [17,35], and other areas with significant ecological characteristics. In contrast, few studies have been conducted on resource cities with a high degree of industrialization and a more serious ecological environment. Therefore, urban ecosystem development can be considered by superimposing the mine resistance surfaces [36]. The minimum cumulative resistance (MCR) model can estimate the energy consumption of species migrating between ecological sources and measure all possible species movement trends [27], which reflect the interactions among landscape patterns and ecological processes [37].Thus, the MCR model has been employed extensively to construct ecological corridors [38,39].

In this study, we applied the InVEST model to the Fushun urban area to evaluate the ecosystem service capacity and determine the ecological source areas using the landscape connectivity index. We then selected ecological corridor resistance factors from natural characteristics, ecological resources, and economic and social aspects. The factor weights were established using AHP to construct a comprehensive ecological resistance surface in Fushun City under the influence of open-pit mining. The MCR model was then used to extract the ecological corridors, which were classified using the gravity model. "Ecological breakpoints" were identified at sites where the ecological corridors crossed mines, expressways, and first-class highways. Combined with the planning of mining and mine rehabilitation in Fushun City, the ecological network optimization strategy proposed here for Fushun City was developed to provide a scientific basis for the reconstruction and optimization of the city's ecological network. The method is universal and can be applied to other resource-oriented cities affected by large open-pit mines.

Starting from the characteristics of damaged urban ecosystems, this study quantitatively assesses the spatial differentiation characteristics of urban ecosystem service functions under the influence of large-scale open-pit mines, and incorporates the quantified ecosystem service functions into the identification and optimization of the ecological network, which is of great significance as a reference for the overall protection of regional ecosystem structure under the influence of large-scale open-pit mining and for the concrete solution of the problem of ecological environment damage.

## Materials and methods

### Study area

Fushun is known as the "Coal Capital" and is located in northeastern Liaoning Province, bordering Shenyang to the west (41˚14´10"–42˚28´32" N, 123˚39´42"–125˚28´58" E). Fushun is located in the middle temperate zone, has a temperate continental monsoon climate, and covers 11271.03 km$^2$, with jurisdiction over four districts and three counties. The urban area is located on the alluvial plain of the Hun River and is surrounded by mountains on three sides. Fushun City has 34 kinds of mineral resources, including metals, non-metals, and coal, with a total of ~5.497 billion tons and total reserves of ~4.332 billion tons. After centuries of mining, the West open-pit mine has become the largest open-pit mine in Asia, which includes the West open-pit slope deformation area (7.03 km$^2$), coal mining subsidence area (18.41 km$^2$), and waste dumps (coal gangue and other waste accumulation; 21.49 km$^2$). Under the joint action of the East and West open-pit mines, coal mining influence areas were formed in the urban areas of the Xinfu, Wanghua, and Dongzhou districts of Fushun City (Fig 1). In recent years, Fushun City has conducted vigorous mine management and ecological restoration and has used the comprehensive remediation and integrated utilization of the West open-pit mine as a breakthrough to propel the growth of the city. In this study, we constructed a potential ecological network for Fushun City, combined ecological restoration of the mining area with urban ecological network development, and proposed scientific recommendations for developing urban ecosystems in Fushun City.

### Data sources

Remote sensing imagery of the Fushun urban area in 2022 acquired by the landsat 8 (http://landsat.visibleearth.nasa.gov/) was used in this study, which has a resolution of 30 m × 30 m. Land cover data for the Fushun urban area in 2022 (30 m × 30 m resolution) were obtained by interpreting the remote sensing images. Digital elevation model data (30 m resolution) were obtained from the Geospatial Data Cloud (https://www.gscloud.cn/), and normalized difference vegetation index data (30 m resolution) were obtained from the Google Earth Engine

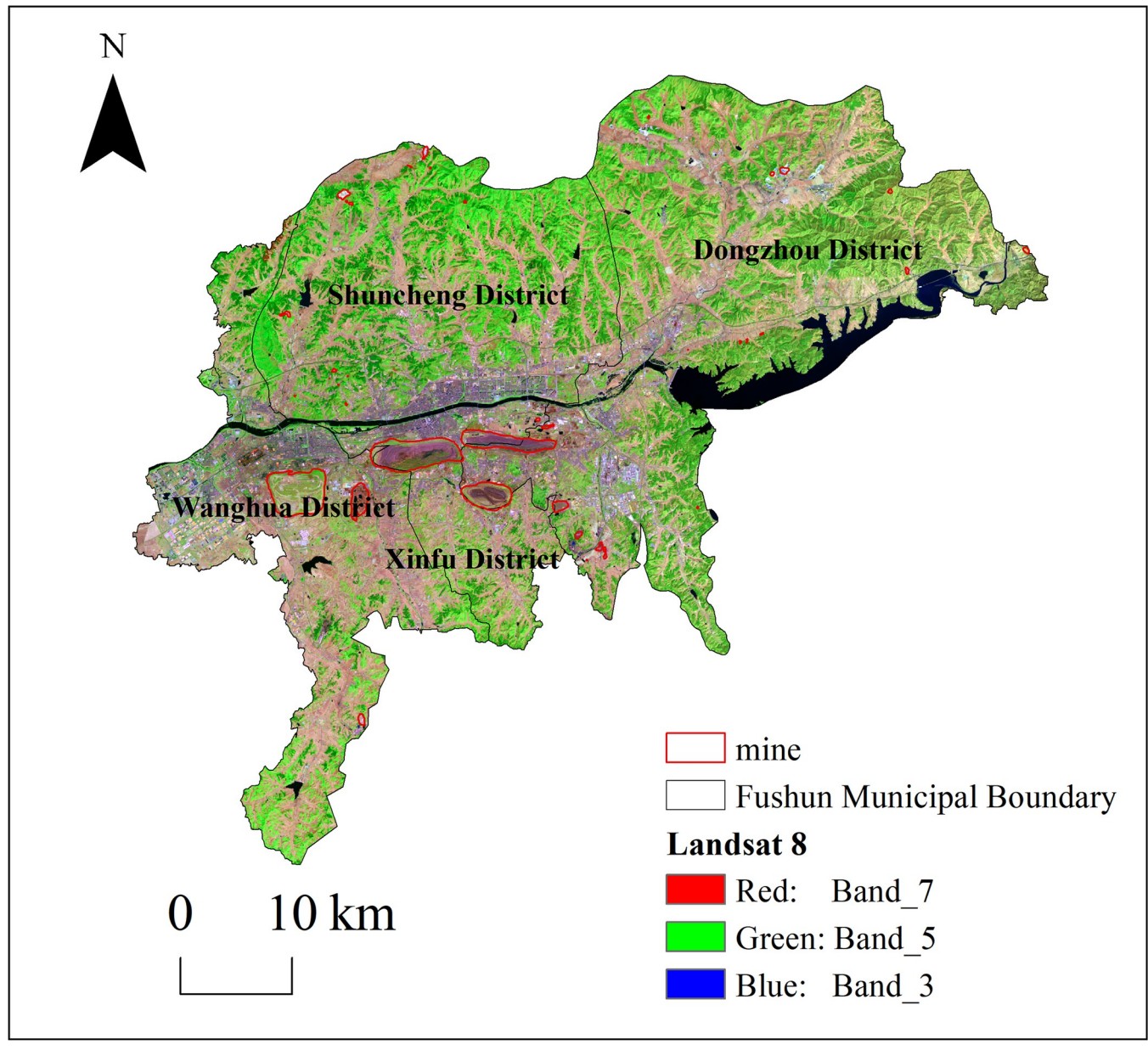

**Fig 1. Remote sensing map of Fushun City, China.**

(https://earthengine.google.com/). Nighttime light data were obtained from the National Oceanic and Atmospheric Administration website. Fushun Municipal Boundary [40] and road data (500 m resolution) were obtained from the Resource and Environment Science Data Center of the Chinese Academy of Sciences (https://wwwresdc.cn/). Mining data were obtained from various mining companies, field surveys, and remote sensing imagery.

Rainfall data for 2022 were obtained from the 1 km resolution China monthly precipitation dataset for 1901–2022 published by the National Earth System Science Data Center (https://www.geodata.cn/). Potential evapotranspiration data for 2022 were obtained from the 1 km resolution China monthly potential evapotranspiration dataset for 1901–2022 published by National Earth System Science Data Center (https://www.geodata.cn/). Soil texture data were

obtained from the SoilGrids dataset, which is a global soil dataset published by the International Soil Reference and Information Center (https://files.isric.org/soilgrids/former/2017-03-10/data). Soil depth data were obtained from the National Cryosphere Desert Data Center. (http://www.ncdc.ac.cn/).

## Research methods

**Modeling urban ecosystem services influenced by open-pit mines.** The habitat quality module of the InVEST model evaluates the ability of the ecological environment to provide suitable living conditions for species [41]. In this study, forests, grassland, and water bodies, all of which have high ecological suitabilities, were used as ecological land, and areas with high impacts from human activity (e.g., mines, construction land, cropland, and bare land) were used as the threat sources [8,42]. The habitat quality was calculated as follows:

$$Q_{xj} = H_j\left(1 - \left(\frac{D_{xj}^z}{D_{xj}^z + k^2}\right)\right) \tag{1}$$

where $Q_{xj}$ denotes the habitat quality corresponding to raster x in habitat type j, $H_j$ is the habitat suitability of the habitat type, $D_{xj}$ is the degree of habitat degradation in raster x, z is the scale constant, and k is the half-saturation constant.

Sediment retention is a crucial regulatory function used to address soil erosion, which is common in mining areas [8]. Sediment retention can be described as the ability of an ecosystem to maintain the soil stability [43]. In this study, the capacity of soil for sand retention services was calculated based on the modified universal soil loss equation algorithm using the InVEST model's sediment delivery ration module, which is calculated as follows:

$$SR_i = \frac{R_i \times K_i \times LS_i(1 - C_i \times P_i)SDR_i}{SDR_{max}} \tag{2}$$

where $SR_i$ is the sediment retention in pixel i (tons/pixel), $R_i$ is the rainfall erosive force, $K_i$ is the soil erodibility factor, $C_i$ is the vegetation cover management factor, $P_i$ is the soil and water conservation factor, $LS_i$ is the slope–length erosion factor, $SDR_i$ is the sand transport ratio in grid i, and $SDR_{max}$ is the maximum theoretical sand transport ratio.

Water yield is a crucial water supply service that is essential for addressing water demand and security [44]. In the context of large-scale open-pit coal mining, water-related studies are essential for understanding water runoff and quality [8]. In this study, water supply services were assessed using the InVEST model's water yield module and were calculated as follows:

$$Y_{xj} = \left(1 - \frac{AET_{xj}}{P_x}\right) \times P_x \tag{3}$$

$$\frac{AET_{xj}}{P_x} = \frac{1 + \omega_x R_{xj}}{1 + \omega_x R_{xj} + \left(\frac{1}{R_{xj}}\right)} \tag{4}$$

where $Y_{xj}$ is the actual evapotranspiration of the land cover type in raster x, $AET_{xj}$ is the actual average annual evapotranspiration of land use type j corresponding to raster x, $P_x$ is the average annual rainfall of raster x, $PET_{xj}$ is the potential evapotranspiration of raster x, $\omega$ is the effective water content of the vegetation, and $R_{xj}$ is the aridity index for the land cover type in raster x [24,45].

Carbon storage and sequestration are important indicators of the scale and amount of primary productivity in an ecosystem [46]. The carbon storage and sequestration in the study

area were calculated based on the distributions of different land-use types and the average carbon density as follows:

$$C_{total} = C_{soil} + C_{above} + C_{below} + C_{dead} \tag{5}$$

where $C_{total}$ is the regional carbon stock; $C_{soil}$ is the soil carbon stock; $C_{below}$ is the underground carbon stock; and $C_{dead}$ is the dead organic matter carbon stock.

The landscape connectivity index can quantify the degree of communication between species and energy in ecological sources. Based on previous studies, the probability of connectivity (PC) and patch importance (dPC) were selected to evaluate the connectivity of the ecological source areas [47,48] as follows:

$$PC = \sum_{i=1}^{n} \sum_{j=1}^{n} \frac{a_i a_j P^*_{ij}}{A_L^2} \tag{6}$$

$$dPC = \frac{PC - PC_{remove}}{PC} \tag{7}$$

where *PC* is the possible connectivity index of the landscape and *dPC* assesses the importance of ecological patches; n denotes the total number of patches; $a_i$ and $a_j$ denote the areas of patches i and j, respectively; $P^*_{ij}$ is the maximum species dispersal probability between patches i and j; $A_L$ is the total area of the landscape; and $PC_{remove}$ denotes the index of possible connectivity in the landscape, excluding patch i.

Ecological sources are large habitat patches that play important roles in regional ecological processes and functions and are crucial for material circulation within regional ecosystems. Based on the habitat quality, sediment retention, water production, and carbon storage calculations, we set the weights of four types of ecosystem service functions. The correlations among the indicators were comprehensively considered when determining the weights, and the results of the four ecosystem service function indicators were normalized by range, divided into 3 km × 3 km grids, and averaged for each grid. The raster calculator in ArcGIS (ver. 10.8) was then used to investigate the ecosystem service functions in the region, which were classified into five levels (i.e., very important, important, average, not important, and very not important) using the natural breakpoint method. The principal components were calculated and the weights were determined using the principal component analysis method in the SPSS (ver. 27.0) software package [49]. Very important and important patches were selected as alternative patches. Using areas of 7, 4, and 1 km² as thresholds, alternative patches were categorized into super-large, large, medium, and small patches. Super-large patches were selected for landscape connectivity analysis, which was performed using the confer software package. Based on previous findings, the confer distance threshold was set to 2500 m and the patch connectivity probability was set to 0.5 [27].

**Ecological resistance surface construction.** The ecological resistance surface refers to the factors in an ecosystem (e.g., topographic features, impacts of human activity, and land cover types) that produce resistance values to the flow of ecological substances among ecological sources [50]. Constructing comprehensive resistance surfaces can better reflect trends and potential ecological spatial processes [22]. Since Fushun is a coal-resource-exhausted city, the influence of mines on the ecological resistance surface cannot be ignored. We selected nine ecological resistance factors (Table 1) from three dimensions: natural endowment, ecological resources, and economic society. Using the Yaahp software package, the relative weight of each resistance element was calculated by comparing its relative importance to other factors. The relative value of the resistance factor was set between 1 and 5. The raster calculator in

**Table 1. Ecological resistance factors and resistance coefficients used in this study.**

| Resistance classification | Resistance factor | Weight | Resistance coefficient | | | | |
|---|---|---|---|---|---|---|---|
| | | | 1 | 2 | 3 | 4 | 5 |
| Natural talent | Slope | 0.11 | ≤10 | (10,20] | (20,30] | (30,40] | >40 |
| | DEM | 0.06 | (-13,125]; (-125,-13] | (125,200]; (-200,-125] | (200,350]; (-326,-200] | (350–500] | >500 |
| Ecological resources | Land cover | 0.25 | Forest | Water body | Grassland | Cropland | Artificial surfaces, mines, bare land |
| | NDVI | 0.08 | >0.6 | (0.5,0.6] | (0.4,0.5] | (0.2–0.4] | ≤0.2 |
| | Distance to water body | 0.11 | ≤500 | (500,1000] | (1000,1500] | (1500,2000] | >2000 |
| Human development | Distance to motorway | 0.16 | >2000 | (1500,2000] | (1000,1500] | (500,1000] | ≤500 |
| | Distance to highway | 0.08 | >800 | (600,800] | (400,600] | (200,400] | ≤200 |
| | Nighttime light index | 0.05 | ≤5 | (5,15] | (15,30] | (30,60] | >60 |
| | Mine | 0.1 | — | — | — | — | — |

DEM: Digital elevation model; NDVI: Normalized difference vegetation index.

ArcGIS was used to weight and sum each single-factor resistance surface to obtain a composite resistance surface.

The different mining states of mineral resources have different degrees of resistance to biodiversity and biological migration near mines [51]. The cost of open-pit mining is lower than that of underground mining; however, the scale and amount of waste material generated by open-pit mining are both large, often occupying a large area of land resources. These conditions can lead to surface ecological environmental deterioration, which can increase the tendency for the land to become desertified. The impacts of mining, beneficiation, smelting, and tailing discharge at metal mines are much higher than those at coal and building material mines, and the ecological impacts of coal mining are larger than those of building material mining [52,53]. In this study, we referred to previous research [36] and constructed the mine resistance surface by investigating the mineral resources in the study area, analyzed the mining boundaries by combining remote sensing imagery with relevant historical mining data and consulting with mine ecological restoration experts, and assigned different ecological resistance values to the different mine types in the study area (Table 2).

**Potential ecological corridor extraction.** Ecological corridors link connecting landscape units in space and improve the overall landscape connectivity in a region by connecting scattered ecological sources. Based on the constructed resistance surface, the MCR model was

**Table 2. Ecological resistance factors and resistance coefficients used in this study.**

| | Underground mining | | Open-pit mining | |
|---|---|---|---|---|
| | Producing mines | Abandoned mines | Producing mines | Abandoned mines |
| Metal mines | (0–1000) = 5 | (0–500) = 5 | (0–3000) = 5 | (0–1000) = 5 |
| | (1000–1500) = 3 | (500–1000) = 3 | (3000–4000) = 3 | (1000–2000) = 3 |
| | (>1500) = 1 | (>1000) = 1 | (>4000) = 1 | (>2000) = 1 |
| Coal mines | (0–1000) = 5 | (0–500) = 5 | (0–2500) = 5 | (0–1000) = 5 |
| | (1000–1500) = 3 | (500–1000) = 3 | (2500–3500) = 3 | (1000–2000) = 3 |
| | (>1500) = 1 | (>1000) = 1 | >3500 = 1 | (>2000) = 1 |
| Building material mines | (0–1000) = 5 | (0–500) = 5 | (0–2000) = 5 | (0–800) = 3 |
| | (1000–2000) = 3 | (500–1000) = 3 | (2000–3000) = 3 | (800–1500) = 3 |
| | (>2000) = 1 | (>1000) = 1 | (>3000) = 1 | (>1500) = 1 |

used to calculate the minimum resistance path between each ecological source point and the target source point based on the resistance surface [41], as follows:

$$MCR = \int_{min} \sum_{j=n}^{i=m}(D_{ij} \times R_i)$$ (8)

where MCR denotes the minimum cumulative resistance value, $\int_{min}$ indicates that the minimum cumulative resistance is proportional to the ecological process, $\Sigma$ denotes the cumulative resistance value between units i and j, $D_{ij}$ is the spatial distance of a species from i to j, and $R_i$ is the resistance value of unit i.

The strengths of the interactions between the ecological sources were evaluated using gravity models, which allowed us to judge the ecological corridors and refine the structure of the ecological network [54], as follows:

$$G_{ab} = \frac{N_a N_b}{D_{ab}^2} = \frac{\left[\frac{1}{P_a \times ln(S_a)}\right]\left[\frac{1}{P_b \times ln(S_b)}\right]}{\left(\frac{L_{ab}}{L_{max}}\right)^2} = \frac{L_{max}^2 ln(S_a) ln(S_b)}{L_{ab}^2 P_a P_b}$$ (9)

where $G_{ab}$ represents the gravitational value between source sites $a$ and $b$; $N_a$ and $N_b$ denote the resistance values of the corridor between source sites $a$ and $b$; $P_a$ and $P_b$ are the path cost values between source sites $a$ and $b$; $L_{ab}$ is the path costal value between source sites $a$ and $b$; and $L_{max}$ is the maximum cumulative resistance value of all corridors in the study area.

**Key ecological node identification.** Ecological nodes are nodes in the regional environment that are critical for the diffusion or migration of organisms. In general, an area with frequent human activity and high traffic flow in the ecological corridor is called an "ecological breakpoint" [55]. The presence of "ecological breakpoints" may lead to the isolation of originally intact habitats, which is unfavorable for biomigration or can threaten the survival of species, and should be emphasized in natural landscapes. In this study, we combined maps of the distributions of expressways, first-class highways, and mines to identify "ecological breakpoints."

## Results

### Ecological source selection

Using the InVEST model, the spatial distributions of the four types of ecosystem services in Fushun City (i.e., habitat quality, sediment retention, water yield, and carbon storage) were calculated (Fig 2A–2D). The spatial differentiation of habitat quality was obvious, and key areas were distributed in the water areas of Duofang Reservoir and the Hunhe River basin, as well as in the forested areas to the north and northeast of Fushun City. Forests can prevent surface soil erosion and are involved in water storage and precipitation. Therefore, high-value areas for sediment retention were located in woodland accumulation areas where the water yield was also higher. Unlike the other ecosystem services, the soil organic carbon content of cropland under anthropogenic activity was high.

The lowest ecosystem service levels were located in urban construction lands and areas affected by large open-pit mines south of the Hunhe River. During open-pit mining, native surface vegetation and topsoil are stripped away. The open pit and waste accumulations near coal mines also cause the occasional spontaneous combustion of coal gangue, and the interiors of the mine pit and shechang (gangue mountains) completely lose their biological productivities, such that the biodiversity, soil and water conservation, and carbon storage levels are almost zero. The construction of roads and the residential land surrounding the mining area

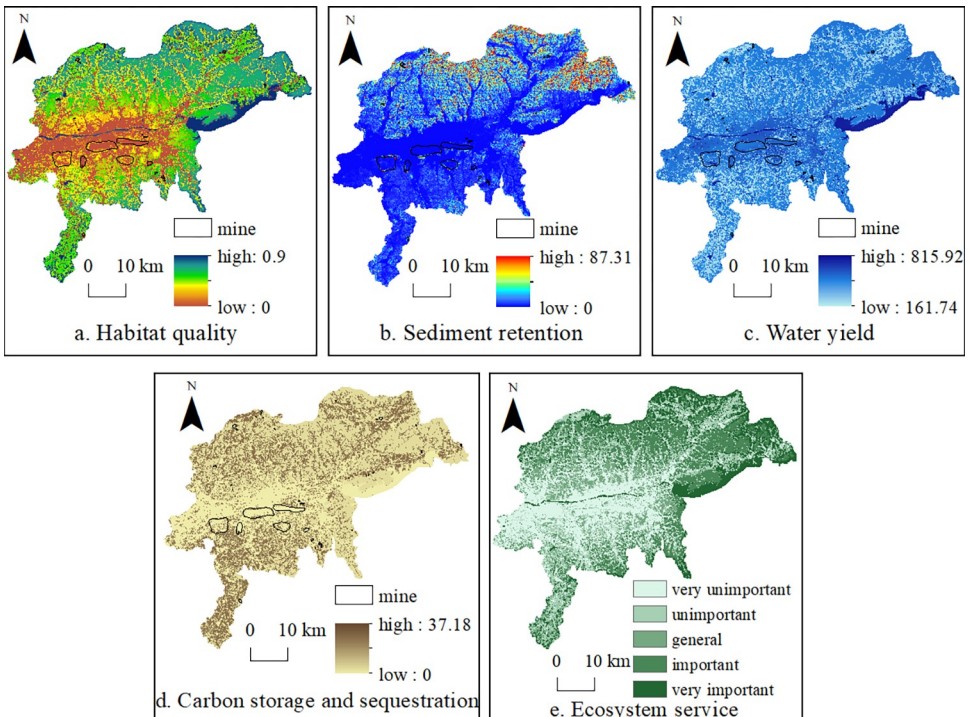

**Fig 2. Spatial distributions of ecosystem services determined using the InVEST model.**

also occupied and destroyed some of the land resources; thus, the overall ecological environment of the open-pit mining impact area has seriously deteriorated. Fig 2 shows that after the recent ecological restoration of open-pit mines, the ecosystem services around the West open pit and restored areas of the east and west shechang have improved, except for the sediment retention capacity. Owing to the unstable geological conditions and steep terrain of the pits and sheds, as well as the long ecological restoration period for soil and water conservation, this effect requires 3–5 years to occur.

The principal component analysis indicated that the KMO value of the four ecosystem service functions was 0.604, and the weights for habitat quality, sediment retention, water yield, and carbon storage were 0.45, 0.24, 0.21, and 0.1, respectively (Table 3 and Fig 2E). Ecological patches were categorized into super-large, large, medium, and small patches, and the numbers and areas of the patches with different sizes were counted. The statistical results indicate that the super-large patches accounted for 51.3% of the total area, large patches accounted for 4.7%,

**Table 3. Landscape connectivity values.**

| Node | Area/km² | dPC | Node | Area/km² | dPC | Node | Area/km² | dPC |
|------|----------|-----|------|----------|-----|------|----------|-----|
| 1 | 79.46 | 39.70 | 8 | 24.18 | 5.78 | 15 | 10.20 | 4.76 |
| 2 | 50.59 | 28.58 | 9 | 15.81 | 0.51 | 16 | 8.95 | 9.33 |
| 3 | 48.94 | 22.94 | 10 | 13.31 | 17.57 | 17 | 8.28 | 3.87 |
| 4 | 41.59 | 23.39 | 11 | 13.03 | 12.56 | 18 | 8.21 | 3.34 |
| 5 | 35.36 | 42.12 | 12 | 10.91 | 13.78 | 19 | 8.01 | 3.33 |
| 6 | 34.08 | 14.77 | 13 | 10.55 | 5.08 | 20 | 7.86 | 7.09 |
| 7 | 34.00 | 21.82 | 14 | 10.47 | 3.89 | 21 | 7.09 | 2.05 |

dPC: Patch importance.

medium patches accounted for 14.5%, and small patches accounted for 29.5%. The landscape fragmentation problem in Fushun City is therefore serious as a result of the influences of mining and human activity. Twenty-one ecological source sites were selected, mainly located in Dongzhou District and the northern Shuncheng District, with no ecological source sites in the impact area of large open-pit mines.

## Ecological resistance surface construction

ArcGIS was used to reclassify each resistance factor, assign grades and values, and weight and stack the values to obtain the base ecological resistance surface, mine resistance surface and integrated ecological resistance surfaces in Fushun City (Figs 3–5).

The ecological resistance surface in Fushun City was higher in the western, central, and southern parts of the region, and lower in the northern and northeastern parts. The resistance value reached a maximum in the East and West open-pit mines and the Shuncheng District building material mining area. Large-scale coal mining not only led to changes in the original land use type, but the deep excavation of the open-pit mines and the high accumulation of coal gangue also produced mining area topography with large undulations, steep terrain, frequent geological disasters, and highly sensitive habitats. In addition, as coal resources and waste must be transported outward, the internal road network in the mining area was relatively rich, which seriously affects the flow of species. Abandoned open-pit mines undergoing ecological management and restoration, resource integration, and reuse are crucial for the ecological development of the city. The resistance value of the urban construction area was only lower than those of mines without ecological restoration, which can be attributed to a larger impervious surface area that hinders ecological processes, such as species migration and material circulation [56]. The low resistance value areas were mainly located in the northern Shuncheng District and the northern and southeastern Dongzhou District, which were less affected by human socioeconomic impacts and had superior natural endowments and rich ecological resources.

## Potential ecological corridor extraction

Based on the ecological sources and comprehensive ecological resistance surface, the cost rasters between the ecological sources and destinations were calculated, and the minimum

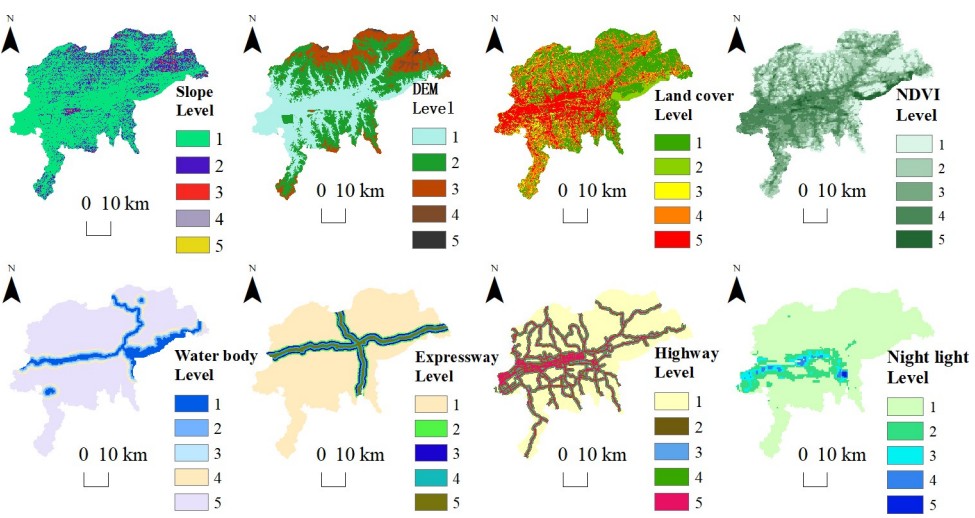

**Fig 3. Base ecological resistance surface in Fushun City.**

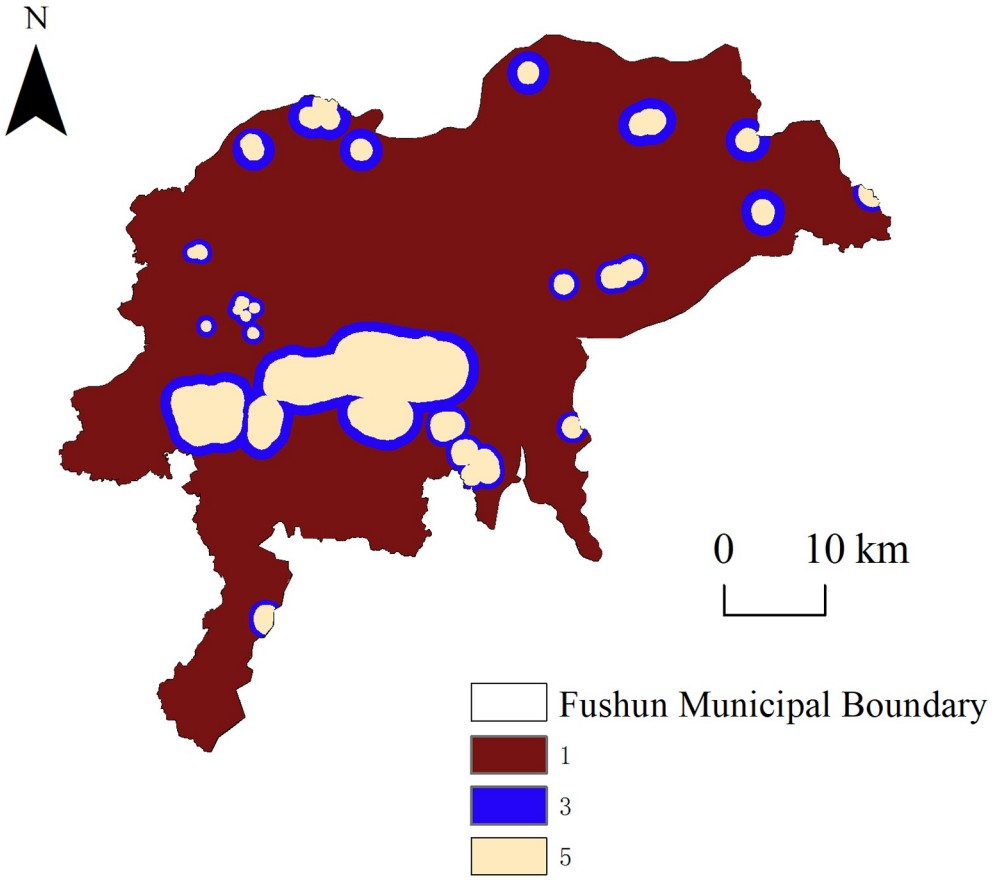

**Fig 4. Mine resistance surface.**

cumulative resistance channel between each source was obtained, which identified 210 ecological corridors. The interaction matrix between the source areas obtained from the gravity model was used, and the mean value of the interaction size (48.24) was set as the threshold value for classifying the ecological corridors. Ecological corridors with interaction sizes greater than the mean value were considered to be important ecological corridors (46), whereas those with values smaller than the mean were considered general ecological corridors (Fig 6). Redundant ecological corridors that passed through the same source area were eliminated.

The interaction matrix (Table 4) indicates that the interaction intensities between ecological source sites No. 7 and No. 19, and between No. 7 and No. 5 were the highest (both >700), followed by the interaction intensity between sites No. 12 and No. 26 (523.95). The geographical space between ecological source sites Nos. 1 and 6 was very close; site No. 6 is rich in forest resources, and site No. 1 is mainly the Dahuofang Reservoir, which is dominated by watersheds. The interaction intensities among site Nos. 9 and 10, 11, 13, and 14 were greater than 10, but were separated from those of Nos. 3, 5, 7, 18, 19, and 21 by large open-pit mines and the main urban area; thus, their interaction intensities were very small (all <10). The lack of ecological sources and the high resistance values in the open-pit mines and the surrounding area caused the number of ecological corridors to be extremely low, and only two general ecological corridors passed through the open-pit mining area.

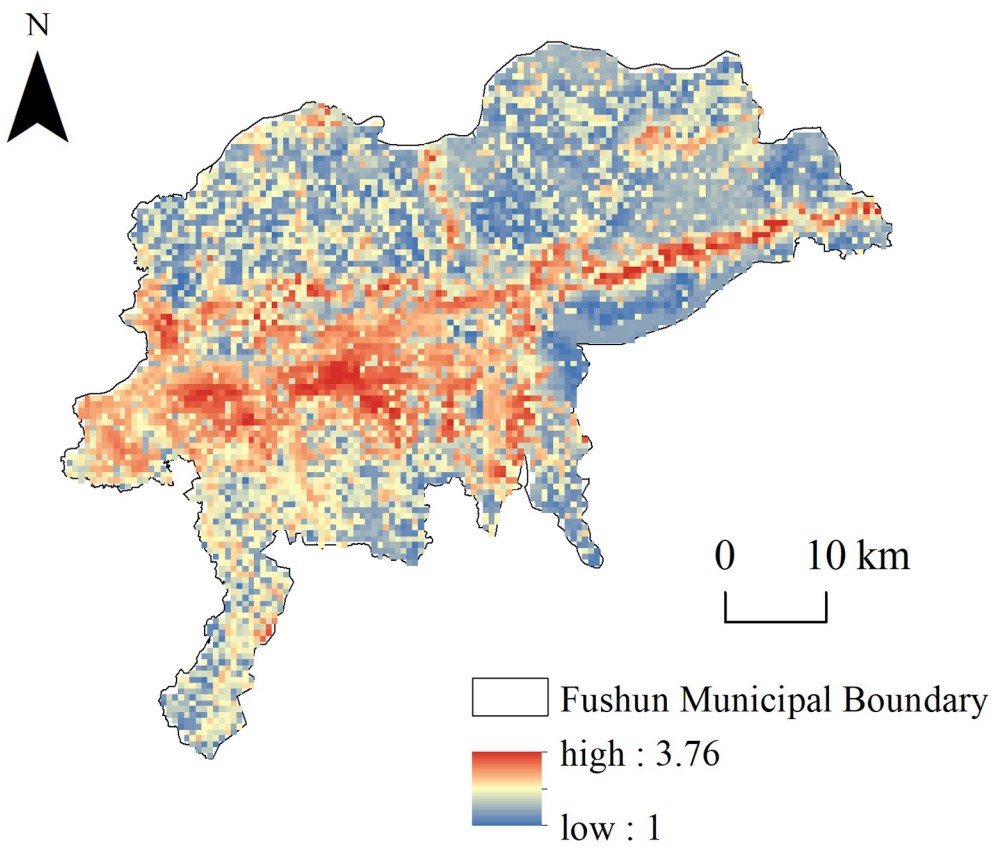

**Fig 5. Integrated ecological resistance surfaces.**

### Key ecological node identification

Based on the constructed ecological corridors, we identified "ecological breakpoints" by combining the expressway, first-class highway, and mine distribution maps. The Middle-Liaoning Ring and Shenyang–Jilin expressways pass through many important ecological corridors, and a first-class highway passes through the city center, intersecting with many general ecological corridors and with many important ecological corridors in the Dongzhou District, forming a total of 37 "ecological breakpoints." Fig 7 shows that the ecological corridor between site Nos. 9 and 21 passes through the West open-pit mine, and the ecological corridors from site No. 9 to Nos. 3 and 19 passes through the western part of the East open-pit mine, thereby forming two "ecological fracture surfaces."

### Ecological network optimization

The central and southern parts of Fushun City lacked ecological sources and had few ecological corridors. Considering the tight land resources in the city center and the wasted land resources influenced by the large-scale open-pit mines, we added ecological sources in the peripheral areas of the city center with high ecosystem service functions. The degree of fragmentation of cultivated land and the landscape near site No. 21 in the northern Shuncheng District was high. Integrating these small land patches can improve the degree of cropland

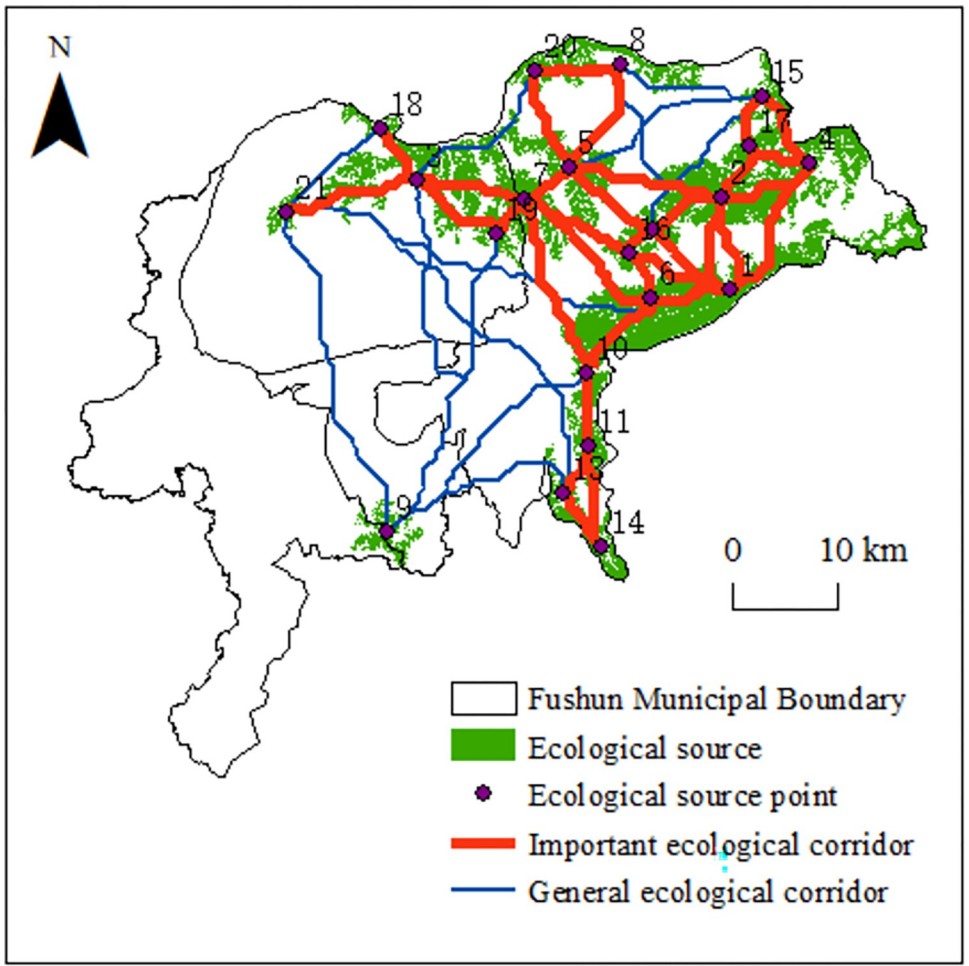

**Fig 6. Ecological network in Fushun City.**

fragmentation and create a new ecological source. Continued ecological management in the Xishechang District that connect it with the surrounding grasslands and plant grass with trees will allow the Wanghua District to form another ecological source. In addition, ecological restoration work is ongoing at the West open-pit mine. If the West open-pit mine is transformed into a large ecological park or a water storage power generation project, it can also become a new ecological source.

Woodlands are suitable for species exchange and diffusion and can better protect species diversity and provide temporary habitats for species movement as corridors. By connecting landscapes, woodlands also have a certain ecological protection function. The study area can directly use existing forested land to upgrade the general ecological corridors. In the central urban area, various green belts and boulevards will be constructed to restore and upgrade ecological corridors, thereby promoting the integrity of the urban ecological network.

The superimposition of ecological corridors and road networks produced 37 "ecological breakpoints," 18 of which were located in important corridors, indicating that important ecological corridors were seriously divided by the road network and hindered the migration and diffusion of species. The breakpoints should be restored and reconstructed, warning signs and speed limits should be established, or special biological migration corridors should be constructed to mitigate these issues. The ecological corridors and mines were superimposed to

**Table 4. Matrix of interaction intensities between the ecological sources.**

| | 1 | 2 | 3 | 4 | 5 | 6 | 7 | 8 | 9 | 10 | 11 | 12 | 13 | 14 | 15 | 16 | 17 | 18 | 19 | 20 | 21 |
|---|---|---|---|---|---|---|---|---|---|---|---|---|---|---|---|---|---|---|---|---|---|
| 1 | 0 | 170.49 | 17.86 | 81.01 | 48.40 | 414.06 | 40.97 | 25.72 | 7.97 | 91.45 | 29.71 | 112.05 | 18.92 | 15.54 | 26.23 | 71.02 | 42.18 | 7.99 | 18.07 | 13.17 | 6.04 |
| 2 | | 0 | 21.94 | 209.00 | 86.27 | 76.44 | 58.65 | 48.92 | 4.94 | 31.59 | 13.14 | 201.05 | 9.52 | 8.35 | 81.70 | 74.12 | 259.99 | 9.39 | 22.60 | 20.18 | 6.23 |
| 3 | | | 0 | 12.60 | 71.69 | 21.15 | 162.06 | 24.69 | 7.33 | 20.63 | 9.12 | 22.56 | 6.92 | 6.20 | 8.69 | 17.36 | 9.15 | 318.40 | 78.46 | 31.56 | 59.19 |
| 4 | | | | 0 | 33.51 | 34.09 | 26.41 | 30.74 | 3.67 | 17.58 | 8.13 | 39.80 | 6.28 | 5.71 | 116.86 | 20.84 | 224.68 | 5.78 | 10.47 | 11.40 | 4.12 |
| 5 | | | | | 0 | 56.52 | 717.70 | 104.90 | 6.30 | 36.46 | 14.34 | 127.90 | 10.20 | 8.80 | 27.98 | 78.24 | 29.99 | 26.13 | 127.22 | 103.88 | 12.90 |
| 6 | | | | | | 0 | 58.02 | 24.62 | 8.79 | 209.05 | 47.04 | 135.70 | 25.68 | 19.66 | 14.68 | 173.96 | 20.59 | 9.00 | 26.14 | 13.67 | 6.73 |
| 7 | | | | | | | 0 | 60.71 | 8.20 | 53.11 | 19.30 | 85.37 | 13.18 | 11.13 | 19.25 | 67.83 | 21.70 | 43.92 | 787.13 | 79.10 | 20.35 |
| 8 | | | | | | | | 0 | 3.29 | 13.82 | 6.42 | 42.27 | 4.98 | 4.53 | 43.16 | 22.85 | 29.66 | 11.99 | 20.20 | 90.02 | 6.14 |
| 9 | | | | | | | | | 0 | 13.32 | 12.86 | 4.27 | 14.05 | 10.03 | 2.03 | 3.68 | 2.12 | 3.40 | 4.83 | 2.62 | 4.32 |
| 10 | | | | | | | | | | 0 | 227.84 | 39.57 | 68.39 | 42.08 | 8.60 | 45.93 | 10.34 | 8.68 | 26.87 | 10.55 | 6.75 |
| 11 | | | | | | | | | | | 0 | 13.90 | 181.97 | 76.85 | 4.11 | 14.08 | 4.72 | 4.08 | 9.70 | 4.76 | 3.29 |
| 12 | | | | | | | | | | | | 0 | 9.21 | 7.71 | 23.11 | 523.95 | 37.70 | 8.91 | 33.48 | 18.84 | 5.76 |
| 13 | | | | | | | | | | | | | 0 | 228.71 | 3.25 | 8.73 | 3.61 | 3.22 | 6.60 | 3.66 | 2.75 |
| 14 | | | | | | | | | | | | | | 0 | 2.99 | 7.08 | 3.27 | 2.94 | 5.57 | 3.30 | 2.46 |
| 15 | | | | | | | | | | | | | | | 0 | 12.64 | 166.78 | 4.38 | 7.47 | 12.38 | 2.75 |
| 16 | | | | | | | | | | | | | | | | 0 | 18.22 | 6.80 | 28.92 | 12.62 | 4.45 |
| 17 | | | | | | | | | | | | | | | | | 0 | 4.04 | 8.21 | 9.57 | 2.78 |
| 18 | | | | | | | | | | | | | | | | | | 0 | 21.19 | 14.29 | 40.81 |
| 19 | | | | | | | | | | | | | | | | | | | 0 | 23.02 | 11.54 |
| 20 | | | | | | | | | | | | | | | | | | | | 0 | 6.16 |
| 21 | | | | | | | | | | | | | | | | | | | | | 0 |

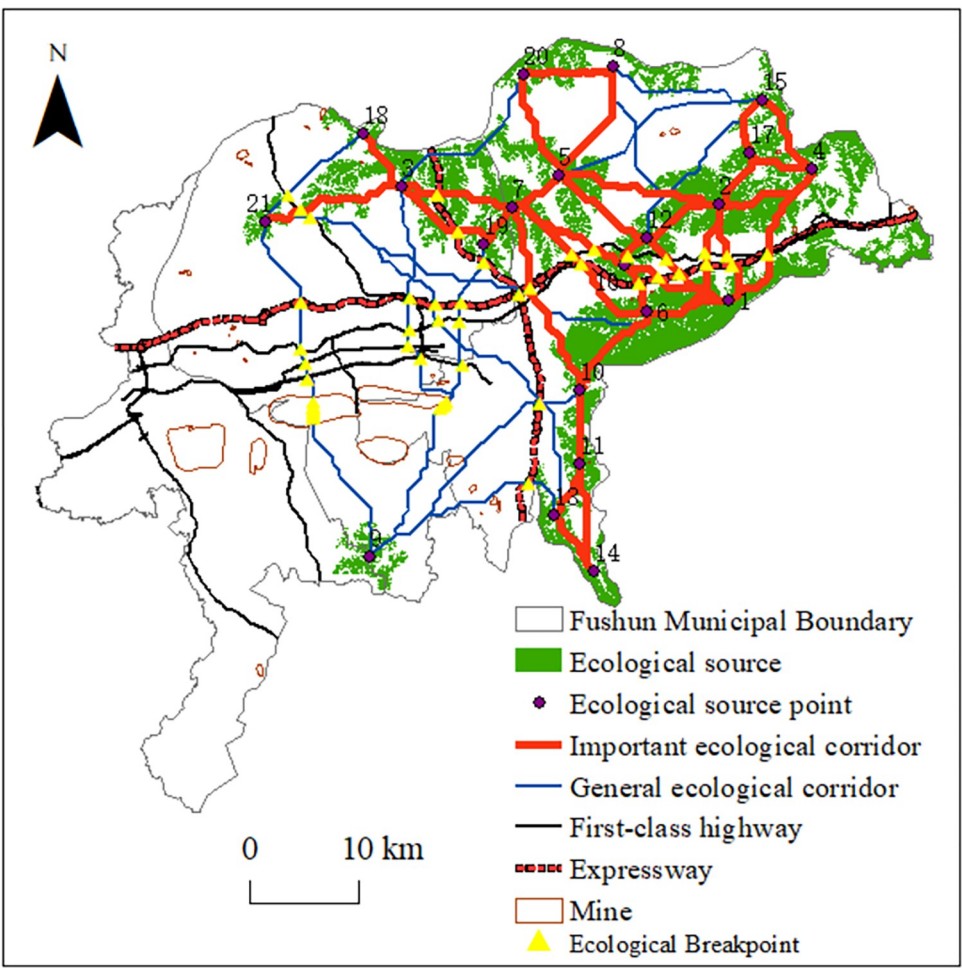

**Fig 7. Ecological breakpoints in Fushun City.**

extract two "ecological fracture surfaces." Although the West open-pit mine is undergoing ecological management and restoration work, owing to the large area and depth of the mining pit, the re-greening project will require a longer period of time for completion. Currently, the southern, western, and northern sides of the West open–pit mine have achieved remarkable results, and biological migration can use the re-greened locations as corridors. Compared with the West open-pit mine, the East open-pit mining area is smaller, shallower, and was mined for a shorter period of time. The current implementation of mining and governance, owing to the expansion of the East open-pit mining work, will move the coal city road to the southern reconstruction. During this process, the ecological corridor to the west of the East open-pit mine will be affected. Therefore, the relationship between ecological environmental protection and mine development and utilization should be properly assessed before beginning the reconstruction and expansion project, and the construction of green mines should be achieved to reduce any damage to ecological corridors or provide alternative solutions.

## Discussion

### Strengthening mine ecological restoration

The construction and optimization of urban ecological networks can improve urban ecosystem protection and promote sustainable urban development. However, the ecosystems in

mining cities are fragile, and large-scale open-pit mining of mineral resources has harmed the natural environment further [57]. A series of geological disasters and environmental problems, including mined-out area breakdowns, surface cracks, landslides, and massive accumulations of coal gangue, have occurred successively. Since they are affected by multiple factors, the construction of ecological networks in mining cities is therefore more difficult than those in other cities.

Strengthening the ecological restoration of open-pit mines is necessary for city development, using the principle of "who mines, who manages, while mining, while managing." With respect to unmanaged open-pit mines, in accordance with the principle of "who manages, who benefits," geological disasters, land damage, vegetation destruction, and other ecological problems can be repaired through manual interventions to stabilize the geological environment, reclaim the damaged land, restore and improve the ecosystem functions [58–60], and motivate reconstruction and optimization of the ecological networks in mining cities.

## Strengths and limitations

The InVEST model is a widely used tool in spatial planning [61], biodiversity conservation [62], ecological compensation [63], and other environmental management decisions [64] at various scales. Ecological sources are important patches that ensure regional ecological security. The InVEST model based on GIS is based on spatial data to quantify multiple ecosystem service functions in the study area and express them in the form of maps, which can realize the spatialization and dynamization of quantitative assessments of ecosystem service functions [65] and increase the scientificity of the source site selection from the perspective of the importance and integrity of ecological patch functions. In the study of ecosystem service functions of mining cities, it can visualize the difference in ecosystem service functions between restored and unrecovered areas of large-scale open-pit mines. The results of this paper are consistent with previous studies that have indicated that mining leads to rapid ecological environmental decline at mines; however, anthropogenic disturbances during the restoration of the mine ecological environment can improve and accelerate recovery [36].

This paper adopts the research paradigm of "ecological source-ecological resistance face-ecological corridor" to study the construction and optimization of urban ecological network under the influence of large open-pit mines, but there are still some limitations, which will be discussed and prospected below.

In the construction of resistance surface, this study selects 9 resistance factors from natural talent, ecological resources, and human development to construct a comprehensive resistance surface in Fushun city. However, the actual study of large open-pit mines is complex and the impact of mining activities on the surrounding ecological environment is also very complex. Moreover, resistance factor selection should be comprehensive and representative; however, how to more rigorously select factors scientifically to reflect the special characteristics of large open-pit mining areas and accurately simulate the ecological network of mining cities require further in-depth study.

As for the extraction results of ecological corridors, this study simplified the corridors into linear elements, and the width of ecological corridors obtained directly by using the MCR model was the pixel width of 30 meters in the grid data, which could not meet the needs of species migration. In fact, the different widths of corridors may have an impact on the ecological mobility, and the degree of connectivity of the network and the heterogeneity of the landscape in reality may also lead to changes in the shape and width of corridors [66]. Therefore, in the next step of the research work, we can consider refining and analyzing the identification of corridor width according to the actual situation of the study area and the results of previous research [67], so as to increase the consideration of corridor restoration.

This study mainly analyzes and discusses the single-year ecological flow of mining cities under the influence of large open-pit mines, and there is not enough research on temporal dynamics. Mineral mining in mining cities often lasts for more than a few decades, and the large open-pit mines in this paper have been mined for a hundred years, but this study has not studied the dynamic changes and dynamic relationships between ecological damage and ecological restoration areas in mining cities. Currently, under the promotion of the management policy of mining ecological environment protection and restoration, new trends of ecological conditions will also occur in large open-pit mines. In the subsequent research, the dynamic simulation and scenario assumptions of the ecological process in the study area can be constructed through the construction of a model to carry out the analysis of the temporal and spatial evolution of ecological flow, with a view to measuring the relationship between ecological environment destruction and restoration triggered by mining exploitation and clarifying the best path for ecological restoration.

## Conclusions

Fushun City exists in China's large open-pit mine-concentrated area. Under the joint action of urban construction and mining, the Fushun City's ecological environment has suffered serious damages and become increasingly sensitive. Therefore, the construction of a complete ecological network planning system to restore the ecological balance of Fushun City is of practical significance. This paper utilizes the InVEST model to assess the ecosystem service function of Fushun City under the influence of large open-pit mines and determines the ecological sources by combining with the landscape connectivity index. Then it constructs a comprehensive ecological resistance surface in Fushun City, extracts ecological corridors in Fushun City using the MCR model, and finally determines ecological breakpoints by combining the distribution characteristics of the mines and the road network, so as to proffer suggestions for the optimization of ecological network management in the Fushun city area.

The results show that, first, critical ecosystem service areas influenced by large open-pit mines in Fushun City are located in the Dongzhou and northern Shuncheng districts, whereas low-value areas are located in construction lands at the city center and the areas affected by open-pit mining south of the Hunhe River. Second, based on a combination of ecosystem service functions and landscape connectivity, 21 ecological sources were identified. Under the influence of large open pit mining, the ecological sources in the study area were unevenly distributed, the internal connectivity of the landscape was weak, and the fragmentation degree was high. Third, the ecological resistance surface in Fushun City generally shows the distribution characteristics of high in the center and west, and low in the surrounding area. High ecological resistance value areas in Fushun City overlapped with mines and expressways. Fourth, the MCR and gravity models identified 210 corridors, including 46 important ecological corridors. The Dongzhou district contained a large number of important ecological corridors, the ecological network structure was complex, and the connectivities between source areas were strong. Few corridors were identified in the mining area, resulting in poor connectivity between the southern and northern parts of Fushun City. Finally, two "ecological fracture surfaces" were extracted under the influences of the west and east open-pit mines. Thirty-seven "ecological breakpoints" were extracted under the influence of the expressway and first-class highway network, 18 of which were located in important ecological corridors.

Based on the findings, Fushun City should strengthen its source protection, strengthen its broken landscape restoration and open-pit mine ecological environmental management/restoration practices, add new ecological sources and corridors, improve the general ecological corridors, and restore and renovate "ecological breakpoints" and "ecological fracture surfaces" in

the region. Enhancing the living environment of the city center and ecological restoration at the mines south of the Hunhe River will improve the network stability and build a safer and more stable ecological network.

## Author Contributions

**Conceptualization:** Dongmei Feng, Ge Bai.

**Data curation:** Ge Bai.

**Formal analysis:** Ge Bai.

**Investigation:** Ge Bai.

**Methodology:** Ge Bai.

**Project administration:** Dongmei Feng.

**Resources:** Liang Wang.

**Software:** Ge Bai.

**Supervision:** Dongmei Feng.

**Validation:** Dongmei Feng, Ge Bai, Liang Wang.

**Visualization:** Ge Bai.

**Writing – original draft:** Ge Bai.

**Writing – review & editing:** Ge Bai, Liang Wang.

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
