## [Decision Letter · Decision Letter 0]

29 Feb 2024

PONE-D-23-35075Influence of large open-pit mines on the construction and optimization of urban ecological networks: A case study of Fushun City, ChinaPLOS ONE

Dear Dr. Bai,

Thank you for submitting your manuscript to PLOS ONE. After careful consideration, we feel that it has merit but does not fully meet PLOS ONE’s publication criteria as it currently stands. Therefore, we invite you to submit a revised version of the manuscript that addresses the points raised during the review process.

We look forward to receiving your revised manuscript.

Kind regards,

Marcela Pagano, Ph.D, M.D.

Academic Editor

PLOS ONE

Journal Requirements:

3. We note that Figures 1-7 in your submission contain map/satellite images which may be copyrighted. All PLOS content is published under the Creative Commons Attribution License (CC BY 4.0), which means that the manuscript, images, and Supporting Information files will be freely available online, and any third party is permitted to access, download, copy, distribute, and use these materials in any way, even commercially, with proper attribution. For these reasons, we cannot publish previously copyrighted maps or satellite images created using proprietary data, such as Google software (Google Maps, Street View, and Earth). For more information, see our copyright guidelines: http://journals.plos.org/plosone/s/licenses-and-copyright.

a. You may seek permission from the original copyright holder of Figures 1-7 to publish the content specifically under the CC BY 4.0 license.  

Reviewers' comments:

Reviewer's Responses to Questions

**Comments to the Author**

1. Is the manuscript technically sound, and do the data support the conclusions?

Reviewer #1: Yes

2. Has the statistical analysis been performed appropriately and rigorously? 

Reviewer #1: Yes

3. Have the authors made all data underlying the findings in their manuscript fully available?

Reviewer #1: Yes

4. Is the manuscript presented in an intelligible fashion and written in standard English?

Reviewer #1: No

5. Review Comments to the Author

Reviewer #1: In recent years, ecological security has been valued, and attention should be paid to both economic development and ecological protection. This study adopts the investment model from the perspective of the impact of mineral resources on ecological security, and uses 22 gravity models to extract and classify ecosystem service capabilities, which has important reference significance. The modification suggestions are as follows:

1.Introduction: Introducing the significance and innovative points of this study.

2.The numbering of the formula should be right aligned.

3.The conclusion should be described in points, and the discussion section should include the shortcomings and prospects of this study.

6. PLOS authors have the option to publish the peer review history of their article (what does this mean?). If published, this will include your full peer review and any attached files.

Reviewer #1: No

---

## [Author Response · Author response to Decision Letter 0]

12 Mar 2024

Dear Editor: 

I wish to re-submit the manuscript titled “Influence of large open-pit mines on the construction and optimization of urban ecological networks: A case study of Fushun City, China.” The manuscript ID is PONE-D-23-35075.

We thank you and the reviewers for your thoughtful suggestions and insights. The manuscript has benefited from these insightful suggestions. I look forward to working with you and the reviewers to move this manuscript closer to publication in the PLOS ONE.

The manuscript has been rechecked and the necessary changes have been made in accordance with the reviewers’ suggestions. The responses to all comments have been prepared and attached herewith. 

Response to "Journal Requirements"：

Question 1. 

Response 1.

We have modified the paper layout according to PLOS ONE's style requirements.

Question 2.

Please note that funding information should not appear in any section or other areas of your manuscript. We will only publish funding information present in the Funding Statement section of the online submission form. Please remove any funding-related text from the manuscript.

Response 2.

We have removed any funding-related text from the manuscript.

Question 3.

We note that Figures 1-7 in your submission contain map/satellite images which may be copyrighted. All PLOS content is published under the Creative Commons Attribution License (CC BY 4.0), which means that the manuscript, images, and Supporting Information files will be freely available online, and any third party is permitted to access, download, copy, distribute, and use these materials in any way, even commercially, with proper attribution. For these reasons, we cannot publish previously copyrighted maps or satellite images created using proprietary data, such as Google software (Google Maps, Street View, and Earth).

Response 3.

We checked all the images (fig.1-7) and found that fig.1 may involve copyright issues as you pointed out, so we have modified fig.1 as a whole. First, we deleted the images that might involve copyright infringement. Then we modified the data source of the remote sensing image, and used the resource "Landsat: http://landsat.visibleearth.nasa.gov/" provided by the journal to update the picture by replacing the elevation data with the remote sensing image data, and changed the remote sensing data source in the manuscript. Finally, the Fushun Municipal boundaries in this paper are not subject to copyright disputes due to the fact that the data were purchased from the data public website as a result of the grant received in the study. According to the requirements of the official website of data download, we have adjusted the data citations in the manuscript and added data citations to the references as references [40]. The related information is as follows:

(1)Direct link to the source of the base map: https://www.resdc.cn/DOI/DOI.aspx?DOIID=120

(2) Map source attribution: Institute of Geographic Sciences and Natural Resources Research, CAS

(3) Map Terms of Use or License Information

“Data Citation

Please state in the data source section: China Multi-year District and County Administrative Boundary Data Data from the Resource and Environmental Science Data Registration and Publication System, and cite the following data papers:

Xu, Xinliang. Multi-year district and county administrative division boundary data in China. Resource and Environmental Science Data Registration and Publication System (http://www.resdc.cn/DOI), 2023.DOI:10.12078/2023010101”

We have deleted the previous Fig. 1 in the submission system and resubmitted the changed Fig. 1 and changed the name of Fig. 1 in the manuscript. The rest of the images in the paper were drawn using ArcGIS (ver.10.8), which are original and do not involve infringement issues.

Finally, The data that support the findings of this study are openly available in public repository "Figshare" : Bai, Ge (2024). Urban ecological networks and large open-pit mines. figshare. Figure. https://doi.org/10.6084/m9.figshare.25390726.v1

Question 4. 

 Please include captions for your Supporting Information files at the end of your manuscript, and update any in-text citations to match accordingly. Please see our Supporting Information guidelines for more information: http://journals.plos.org/plosone/s/supporting-information.

Response 4.

We uploaded the "other" file to "supporting information" in the last submission, but there is actually no supporting information, we have corrected it in the submission system.

Question 5. 

Please review your reference list to ensure that it is complete and correct.

Response 5.

We have checked all references, and there are no retracted papers.

Response to "Comments to the Author"：

Question 1. 

Introduction: Introducing the significance and innovative points of this study.

Response 1.

We have added the significance and innovative points of this study in the manuscript:

Starting from the characteristics of damaged urban ecosystems, this study quantitatively assesses the spatial differentiation characteristics of urban ecosystem service functions under the influence of large-scale open-pit mines, and incorporates the quantified ecosystem service functions into the identification and optimization of the ecological network, which is of great significance as a reference for the overall protection of regional ecosystem structure under the influence of large-scale open-pit mining and for the concrete solution of the problem of ecological environment damage.

Question 2. 

The numbering of the formula should be right aligned.

Response 2.

We have right-justified the numbers of all formulas.

Question 3. 

The conclusion should be described in points, and the discussion section should include the shortcomings and prospects of this study.

Response 3.

We have revised the manuscript as below to address these. 

(1) We have added a discussion section that includes the shortcomings and prospects of this study.

This paper adopts the research paradigm of “ecological source-ecological resistance face-ecological corridor” to study the construction and optimization of urban ecological network under the influence of large open-pit mines, but there are still some limitations, which will be discussed and prospected below.

“This paper adopts the research paradigm of “ecological source-ecological resistance face-ecological corridor” to study the construction and optimization of urban ecological network under the influence of large open-pit mines, but there are still some limitations, which will be discussed and prospected below.

In the construction of resistance surface, this study selects 9 resistance factors from natural talent, ecological resources, and human development to construct a comprehensive resistance surface in Fushun city. However, the actual study of large open-pit mines is complex and the impact of mining activities on the surrounding ecological environment is also very complex. Moreover, resistance factor selection should be comprehensive and representative; however, how to more rigorously select factors scientifically to reflect the special characteristics of large open-pit mining areas and accurately simulate the ecological network of mining cities require further in-depth study.

As for the extraction results of ecological corridors, this study simplified the corridors into linear elements, and the width of ecological corridors obtained directly by using the MCR model was the pixel width of 30 meters in the grid data, which could not meet the needs of species migration. In fact, the different widths of corridors may have an impact on the ecological mobility, and the degree of connectivity of the network and the heterogeneity of the landscape in reality may also lead to changes in the shape and width of corridors [68]. Therefore, in the next step of the research work, we can consider refining and analyzing the identification of corridor width according to the actual situation of the study area and the results of previous research, so as to increase the consideration of corridor restoration.

This study mainly analyzes and discusses the single-year ecological flow of mining cities under the influence of large open-pit mines, and there is not enough research on temporal dynamics. Mineral mining in mining cities often lasts for more than a few decades, and the large open-pit mines in this paper have been mined for a hundred years, but this study has not studied the dynamic changes and dynamic relationships between ecological damage and ecological restoration areas in mining cities. Currently, under the promotion of the management policy of mining ecological environment protection and restoration, new trends of ecological conditions will also occur in large open-pit mines. In the subsequent research, the dynamic simulation and scenario assumptions of the ecological process in the study area can be constructed through the construction of a model to carry out the analysis of the temporal and spatial evolution of ecological flow, with a view to measuring the relationship between ecological environment destruction and restoration triggered by mining exploitation and clarifying the best path for ecological restoration”.

(2) Moreover, we have presented the conclusion in points

Fushun City exists in China’s large open-pit mine-concentrated area. Under the joint action of urban construction and mining, the Fushun City’s ecological environment has suffered serious damages and become increasingly sensitive. Therefore, the construction of a complete ecological network planning system to restore the ecological balance of Fushun City is of practical significance. This paper utilizes the InVEST model to assess the ecosystem service function of Fushun City under the influence of large open-pit mines and determines the ecological sources by combining with the landscape connectivity index. Then it constructs a comprehensive ecological resistance surface in Fushun City, extracts ecological corridors in Fushun City using the MCR model, and finally determines ecological breakpoints by combining the distribution characteristics of the mines and the road network, so as to proffer suggestions for the optimization of ecological network management in the Fushun city area.

The results show that, first, critical ecosystem service areas influenced by large open-pit mines in Fushun City are located in the Dongzhou and northern Shuncheng districts, whereas low-value areas are located in construction lands at the city center and the areas affected by open-pit mining south of the Hunhe River. Second, based on a combination of ecosystem service functions and landscape connectivity, 21 ecological sources were identified. Under the influence of large open pit mining, the ecological sources in the study area were unevenly distributed, the internal connectivity of the landscape was weak, and the fragmentation degree was high. Third, the ecological resistance surface in Fushun City generally shows the distribution characteristics of high in the center and west, and low in the surrounding area. High ecological resistance value areas in Fushun City overlapped with mines and expressways. Fourth, the MCR and gravity models identified 210 corridors, including 46 important ecological corridors. The Dongzhou district contained a large number of important ecological corridors, the ecological network structure was complex, and the connectivities between source areas were strong. Few corridors were identified in the mining area, resulting in poor connectivity between the southern and northern parts of Fushun City. Finally, two “ecological fracture surfaces” were extracted under the influences of the west and east open-pit mines. Thirty-seven “ecological breakpoints” were extracted under the influence of the expressway and first-class highway network, 18 of which were located in important ecological corridors. 

Based on the findings, Fushun City should strengthen its source protection, strengthen its broken landscape restoration and open-pit mine ecological environmental management/restoration practices, add new ecological sources and corridors, improve the general ecological corridors, and restore and renovate “ecological breakpoints” and “ecological fracture surfaces” in the region. Enhancing the living environment of the city center and ecological restoration at the mines south of the Hunhe River will improve the network stability and build a safer and more stable ecological network.

Thank you for your consideration. I look forward to hearing from you.

Sincerely,

Ge Bai

College of Business Administration

Liaoning Technical University

[Liaoning Technical University, 188 Longwan South Street, Xingcheng City, Huludao City, Liaoning Province, China, 125100]

Tel.: +17335833151

2433293485@qq.com

---

## [Editor Report · Decision Letter 1]

4 Apr 2024

PONE-D-23-35075R1Influence of large open-pit mines on the construction and optimization of urban ecological networks: A case study of Fushun City, ChinaPLOS ONE

Dear Dr. Bai

Thank you for submitting your manuscript to PLOS ONE. After careful consideration, we feel that it has merit but does not fully meet PLOS ONE’s publication criteria as it currently stands. Therefore, we invite you to submit a revised version of the manuscript that addresses the points raised during the review process.

We look forward to receiving your revised manuscript.

Kind regards,

Marcela Pagano, Ph.D, M.D.

Academic Editor

PLOS ONE

Journal Requirements:

Additional Editor Comments:

Dear authors, please, check for references not found in the revision version R1,

---

## [Author Response · Author response to Decision Letter 1]

4 Apr 2024

[April 5, 2024]

Dear Editor: 

I wish to re-submit the manuscript titled “Influence of large open-pit mines on the construction and optimization of urban ecological networks: A case study of Fushun City, China.” The manuscript ID is PONE-D-23-35075R1.

We thank you and the reviewers for your thoughtful suggestions and insights. The manuscript has benefited from these insightful suggestions. I look forward to working with you and the reviewers to move this manuscript closer to publication in the PLOS ONE.

The manuscript has been rechecked and the necessary changes have been made in accordance with the reviewers’ suggestions. The responses to all comments have been prepared and attached herewith. 

Question about "Journal Requirements": Please review your reference list to ensure that it is complete and correct.

Question about "Additional Editor Comments": please, check for references not found in the revision version R1.

Response to the above questions:

Due to our citation of references in the process of the wrong operation led to the citation reported error, we have made changes to the citation of references, and there is no withdrawn paper.

Along with the modification of the submission, we uploaded the figure files to the Preflight Analysis and Conversion Engine (PACE) digital diagnostic.

Finally, The data that support the findings of this study are openly available in public repository "Figshare" at: https://doi.org/10.6084/m9.figshare.25390726.v1.

Thank you for your consideration. I look forward to hearing from you.

Sincerely,

Ge Bai

College of Business Administration

Liaoning Technical University

[Liaoning Technical University, 188 Longwan South Street, Xingcheng City, Huludao City, Liaoning Province, China, 125100]

Tel.: +17335833151

2433293485@qq.com

---

## [Editor Report · Decision Letter 2]

18 Apr 2024

Influence of large open-pit mines on the construction and optimization of urban ecological networks: A case study of Fushun City, China

PONE-D-23-35075R2

Dear Dr. Ge Bai,

We’re pleased to inform you that your manuscript has been judged scientifically suitable for publication and will be formally accepted for publication once it meets all outstanding technical requirements.

Kind regards,

Marcela Pagano, Ph.D, M.D.

Academic Editor

PLOS ONE
---

## [Editor Report · Acceptance letter]

26 Apr 2024

PONE-D-23-35075R2 

PLOS ONE

Dear Dr. Bai, 

I'm pleased to inform you that your manuscript has been deemed suitable for publication in PLOS ONE. Congratulations! Your manuscript is now being handed over to our production team.

Kind regards, 

on behalf of

Dr. Marcela Pagano 

Academic Editor

PLOS ONE